# Adverse Pregnancy Outcomes and Maternal Chronic Diseases in the Future: A Cross-Sectional Study Using KoGES-HEXA Data

**DOI:** 10.3390/jcm11051457

**Published:** 2022-03-07

**Authors:** Geum Joon Cho, Jiae Kim, Ji Young Kim, Sung Won Han, Soo Bin Lee, Min-Jeong Oh, Sa Jin Kim, Jae Eun Shin

**Affiliations:** 1Department of Obstetrics and Gynecology, College of Medicine, Korea University, Seoul 02841, Korea; md_cho@hanmail.net (G.J.C.); mjohmd@korea.ac.kr (M.-J.O.); 2Department of Obstetrics and Gynecology, College of Medicine, The Catholic University of Korea, Seoul 06591, Korea; kja3231@nate.com (J.K.); gabrielle83@naver.com (J.Y.K.); ksajin@catholic.ac.kr (S.J.K.); 3School of Industrial Management Engineering, Korea University, Seoul 02841, Korea; swhan@korea.ac.kr (S.W.H.); log0629@korea.ac.kr (S.B.L.)

**Keywords:** metabolic syndrome, preeclampsia, gestational diabetes mellitus, low birth weight, macrosomia, cardiovascular disease

## Abstract

Adverse pregnancy outcomes (APOs) are associated with an increased risk of chronic diseases, including cardiovascular disease (CVD) and metabolic syndrome (MS), in the future. We designed a large-scale cohort study to evaluate the influence of APOs (preeclampsia, gestational diabetes mellitus (GDM), stillbirth, macrosomia, and low birth weight) on the incidence of chronic diseases, body measurements, and serum biochemistry in the future and investigate whether combinations of APOs had additive effects on chronic diseases. We used health examinee data from the Korean Genome and Epidemiology Study (KoGES-HEXA) and extracted data of parous women (*n* = 30,174; mean age, 53.02 years) for the analysis. Women with APOs were more frequently diagnosed with chronic diseases and had a family history of chronic diseases compared with women without APOs. Composite APOs were associated with an increased risk of hypertension, diabetes mellitus, hyperlipidemia, angina pectoris, stroke, and MS (adjusted odds ratio: 1.093, 1.379, 1.269, 1.351, 1.414, and 1.104, respectively) after adjustment for family history and social behaviors. Preeclampsia and GDM were associated with an increased risk of some chronic diseases; however, the combination of preeclampsia and GDM did not have an additive effect on the risk. APOs moderately influenced the future development of maternal CVD and metabolic derangements, independent of family history and social behaviors.

## 1. Introduction

Chronic diseases, including cardiovascular disease (CVD) and its risk factors such as diabetes mellitus and metabolic syndrome (MS), are now recognized as the leading factors that threaten women’s health worldwide. The overall prevalence and cumulative incidence of CVD per 1000 individuals were higher in women than in men in 2015 (105.62 vs. 96.52 and 71.93 vs. 62.15, respectively) [1]. Prevalence rates of both coronary heart disease (CHD) and stroke have risen to similar levels as those of men of a similar age [2,3]. CHD mortality rates have increased in American women within the age range of 35–54 years, contrary to a decreasing trend in the past four decades from 1980 to 2000 [2,4,5]. Additionally, in Korea, the CVD mortality rate is 1.1 times higher in women than that in men [6]. These suggest that underlying factors leading to CVD may be different in women compared with men, and identification of the pathophysiology unique to women has considerable potential to improve the long-term health status of women.

With an improved understanding of cardiovascular changes during normal pregnancy and the changes that are associated with adverse pregnancy outcomes (APOs), it is now recognized that obstetric events can contribute to gender-specific differences in cardiovascular risk and the development of CVD later in life [7]. APOs have also been linked to MS, a major risk factor for CVD [8,9]. Guidelines for the prevention of CVD in women include a history of APOs, such as preeclampsia, gestational diabetes mellitus (GDM), preterm delivery, pregnancy-induced hypertension, and delivery of a low birth weight (LBW) infant, as major CVD risk factors [2,10,11,12]. The link between APOs and future chronic diseases mainly focuses on preeclampsia and GDM, which complicate 5 to 10% and 7 to 10% of pregnancies worldwide, respectively [13,14]. Many epidemiologic studies and meta-analyses have shown that preeclampsia is related to an increased risk of developing or dying from overall CVD [15,16,17]. Moreover, preeclampsia is positively associated with cardiovascular risk factors such as hypertension, diabetes mellitus, and MS [15,18,19,20]. GDM is also associated with a significant increase in the risk factors for CVD, including diabetes mellitus and hypertension [21,22,23]. However, other APOs, such as macrosomia, LBW, and stillbirth, have not been extensively studied. Furthermore, whether APOs cause various chronic diseases and affect serum biochemistry in later life and whether a combination of APOs confers higher risks of chronic diseases need to be investigated. Additionally, many studies have failed to adjust for factors such as social behaviors and family history of cardiometabolic status, thus limiting the understanding of CVD risk trajectory.

We, therefore, designed a large-scale cohort study to (1) evaluate the influence of APOs (preeclampsia, GDM, stillbirth, macrosomia, LBW) on the incidence of chronic diseases, body measurements, and serum biochemistry in later life, (2) assess whether APOs were independent risk factors for chronic diseases, and (3) investigate the additive effects of combinations of APOs on chronic diseases.

## 2. Materials and Methods

### 2.1. Study Population and Design

This cohort study relied on data from the Korean Genome and Epidemiology Study (KoGES). The KoGES is a national survey conducted by the Korean government (National Research Institute of Health, Centers for Disease Control and Prevention, and Ministry for Health and Welfare, Republic of Korea) to investigate the genetic and environmental etiology of common complex diseases in Koreans and causes of death. Details of the KoGES and the methods used were comprehensively explored [24]. Among the KoGES Consortium data, we used KoGES health examinee (HEXA) data consisting of data from participants older than 40 years who resided in urban areas. The HEXA cohort included participants from examination centers/medical institutions of 14 major cities across Korea who were recruited between 2004 and 2013. The participating centers were chosen after strict evaluation of community representation and adherence to the inclusion criteria. For baseline recruitment, eligible participants were asked to volunteer through on-site invitation, telephone calls, mailed letters, media campaigns, or community leader-mediated conferences. The detailed inclusion criteria for recruitment are described in a previous study [24]. All recruited participants voluntarily signed an informed consent form and attended follow-up from 2012 to 2016. This study was approved by the Ethics Committee of Korea University (2021GR0007).

### 2.2. Participant Selection

Among 113,945 women, those who lacked records of pregnancy outcome histories (*n* = 25,803), who lacked records of previous medical history, including family history (*n* = 57,808), and with a history of CVD before pregnancy (*n* = 139) were excluded. Finally, 30,174 participants were included in the study.

### 2.3. Survey

The HEXA consisted of two components: a health interview and a health examination. The health interview included questions regarding demographics, lifestyle status, medical history, and pregnancy outcomes. Each participant was asked regarding the previous history of pregnancy outcomes, including preeclampsia, GDM, stillbirth, macrosomia (birth weight of the neonate ≥ 4 kg), and LBW (birth weight of the neonate ≤ 2.5 kg), by trained interviewers. The questions were structured as follows: “Have you ever been diagnosed with preeclampsia?” and “Have you given birth to a baby weighing more than 4 kg?” All questions required a “yes” or “no” answer. Each participant was also asked regarding the medical history of chronic diseases (hypertension, diabetes mellitus, hyperlipidemia, stroke, transient cerebral ischemia, angina pectoris, and Parkinson’s disease) and family history of chronic diseases in siblings or parents. Data for the following covariates were also obtained: age, age at delivery, lactation history, alcohol consumption, and smoking status. Smoking status was divided into two categories: current smoker and non-smoker. Drinking status was also classified into two groups: current drinker and non-drinker.

The health examination included anthropometric measurements and serum laboratory tests. All anthropometric measurements were obtained by trained and skilled examiners using a consistent and standardized methodology. Height was measured to the nearest 1 mm using a portable extensimeter (SECA 225; SECA, Hamburg, Germany). Weight was measured to the nearest 0.1 kg using a calibrated balance scale (GL-6000-20; CAS KOREA, Seoul, Korea). Waist circumference (WC) was measured at the narrowest point between the lower border of the rib cage and the iliac crest during minimal respiration. Hip circumference (HC) was measured at the widest part of the hip in the horizontal plane. Blood pressure (BP) was measured using a standard mercury sphygmomanometer.

All blood samples were obtained after the participants had fasted for a minimum of eight hours. The levels of fasting blood sugar (FBS), γ-glutamyl transferase, aspartate aminotransferase, alanine aminotransferase, albumin, blood urea nitrogen, creatinine, uric acid, total cholesterol, high-density lipoprotein cholesterol (HDL-C), triglycerides, and high-sensitivity C-reactive protein were measured using enzymatic methods.

### 2.4. Covariates

Composite APOs were defined if any of the APOs were present. MS was defined using the criteria proposed by the Joint Interim Statement of the International Diabetes Federation Task Force on Epidemiology and Prevention [25]. MS was defined as the presence of at least three of the five metabolic components described as follows: (i) a WC ≥ 80 cm, according to the International Diabetes Federation criteria for Asian countries; (ii) FBS ≥ 100 mg/dL or treatment for elevated glucose; (iii) serum fasting triglyceride ≥ 150 mg/dL (1.7 mmol/L) or medication use; (iv) serum HDL-C < 50 mg/dL (1.3 mmol/L) or medication use; (v) systolic BP ≥ 130 mmHg, diastolic BP ≥ 85 mmHg, or drug treatment for hypertension.

### 2.5. Statistical Analysis

Continuous and categorical variables are expressed as mean ± standard deviation and percentages, respectively. Basic characteristics, body measurements, and serum biochemistry of the study population were compared between women with APOs and women without APOs, using t-test for continuous variables and chi-square test for categorical variables. Multivariate logistic regression analysis was used to estimate the adjusted odds ratios (aORs) and 95% confidence intervals (CIs) for the development of chronic diseases. For multivariable analyses, we adjusted for maternal age, family history (hypertension, diabetes, angina, and stroke), present smoking, and present alcohol consumption. The significance level was defined as a p-value less than 0.05. Statistical analyses were performed using SAS for Windows (version 9.4; SAS Inc., Cary, NC, USA).

## 3. Results

APOs occurred in 5264 women in the study population (prevalence rate, 17.4%), and the mean age of the participants was 53.02 years. The characteristics of the study participants are listed in Table 1. Women with APOs had lower age (51.70 ± 7.68 vs. 53.42 ± 8.08 years, respectively, *p* < 0.001) but had higher age at first delivery (25.81 ± 3.00 vs. 25.32 ± 3.01 years, respectively, *p* < 0.001) and last delivery (30.22 ± 3.84 vs. 29.75 ± 3.67 years, respectively, *p* < 0.001) than women without APOs. Compared with those without APOs, those with APOs were more frequently diagnosed with some chronic diseases such as diabetes mellitus and hypercholesterolemia and had a family history of chronic diseases, including hypertension, diabetes mellitus, angina pectoris, and stroke. They were also more likely to smoke and drink alcohol and less likely to breastfeed. Prevalence rates of MS and its components were not statistically different between the two groups, except in women with high FBS levels.

Body measurements and laboratory data were also compared between the two groups (Table 2). Women with APOs had greater height, weight, WC, and HC than women without APOs. Mean values of FBS and serum albumin levels were statistically different between the two groups.

Table 3 presents the results of the multiple regression analysis from the model used to investigate the relationship between previous APOs and chronic diseases. After adjusting for confounding factors, women with composite APOs showed higher rates of hypertension (aOR, 1.093; 95% CI, 1.004–1.189), diabetes mellitus (aOR, 1.379; 95% CI, 1.207–1.575), hyperlipidemia (aOR, 1.269; 95% CI, 1.126–1.429), stroke (aOR, 1.414; 95% CI, 1.083–1.847), and angina pectoris (aOR, 1.351; 95% CI, 1.090–1.675) than those without APOs. Transient cerebral ischemia was not associated with composite APOs. Women with preeclampsia only had a higher risk of hypertension, hyperlipidemia, and angina pectoris. Women with GDM only had a higher risk of hypertension, diabetes mellitus, and hyperlipidemia. The risk estimate of GDM for future diabetes mellitus was the highest (aOR, 9.069; 95% CI, 5.938–13.851). In women with a combination of preeclampsia and GDM, the association was significant only with diabetes mellitus (aOR, 3.287; 95% CI, 1.473–7.338), and the association was not additive. Maternal risk of chronic diseases was not associated with stillbirth or LBW among the APOs. Macrosomia was associated with a higher risk of diabetes mellitus and hyperlipidemia, a trend that was similar to that of GDM.

We analyzed the relationship between APOs and future MS and its components (Table 4). Adjusted analysis showed that composite APOs were associated with an increased risk of MS (aOR, 1.104; 95% CI, 1.020–1.194), elevated WC (aOR, 1.172; 95% CI, 1.094–1.255), hypertension (aOR, 1.074; 95% CI, 1.006–1.147), and elevated FBS level (aOR, 1.193; 95% CI, 1.111–1.280). Low HDL and high triglyceride levels were not associated with composite APOs. The effect of preeclampsia only on MS was not statistically significant; however, the effects of GDM only and the combination of preeclampsia and GDM on MS were statistically significant (aOR, 1.908 and 1.866, respectively). The combination of APOs reduced the risk of MS rather than increasing it.

## 4. Discussion

In this large cohort study, an association was found between APOs and subsequent risk of CVD and other cardiovascular risk factors, including MS, after adjusting for family history of chronic diseases and social behaviors. The risk of CVD and cardiovascular risk factors did not increase further when preeclampsia occurred in combination with GDM.

Similar to previous studies, our study showed that women with APOs were associated with a higher risk of CVD as well as cardiovascular risk factors, including MS. Moreover, women with APOs had a higher prevalence of CVD in the family, which is known as an important risk factor for developing CVD [26,27,28]. Although family history plays an important role in the development of CVD, a few large cohort studies have adjusted for family history to assess the correlation between APOs and the occurrence of CVD in later life. The increased risk of CVD attributable to family history can be caused by shared genetic, environmental, and behavioral factors [26]. The association between APOs and CVD remaining after adjustment for family history may also be affected, in part, by maternal environmental and behavioral factors, which could be modifiable. In this study, women with APOs tended to be more obese and had bad habits for cardiometabolic health, especially smoking, alcohol consumption, and less frequent breastfeeding. A previous study also reported that people with a family history of CVD had less favorable risk factors, including physical inactivity and hypercholesterolemia [26]. In this study, we observed a modest but independent association between APOs and the occurrence of CVD in the future after adjusting for family history and social behaviors. Therefore, pregnancy could be a new opportunity to inform women at risk in advance of danger signals and to prevent the occurrence of CVD through lifestyle modifications.

Among the various APOs, preeclampsia and GDM are the most studied to clarify the relationship with chronic diseases. During pregnancy, GDM increases the risk of preeclampsia [29]. However, the combined effect of preeclampsia and GDM on maternal chronic diseases after delivery has not been extensively studied. Only two studies have investigated the combined effects of GDM and preeclampsia on chronic diseases. One study in Canada reported that having both APOs was associated with more than an additive effect on diabetes mellitus, hypertension, and CVD when compared with having neither GDM nor preeclampsia after six months of delivery [30]. Furthermore, a recent study with a small population that was conducted in Turkey also proved that patients with a combination of GDM and preeclampsia had a higher prevalence of CHD than patients with only GDM [31]. Unlike these previous studies, the present study failed to prove the additive effect of the combination of GDM and preeclampsia on chronic diseases. This may be due to differences in the study population and the duration of the follow-up period. One of these previous studies [30] had a limitation in that it was comprised of women with various ethnocultural backgrounds, and the other study [31] had the limitation of a small sample size. Moreover, these studies had a shorter follow-up period of CVD evaluation after delivery than that in the present study. To the best of our knowledge, the present study is the only study that has examined the long-term effect of the combination of preeclampsia and GDM on chronic diseases in a large sample. This finding warrants further analysis as well as a deeper investigation of the possible causes and/or involved mechanisms.

Research regarding the long-term health status of women who deliver infants with LBW or macrosomia or have a history of stillbirth is scarce. We observed that macrosomia was associated with an increased risk of some chronic diseases. This observation may be related to the strong association between macrosomia and GDM as both groups showed similar results in this study. A previous nationwide Swedish study also reported similar results that the risk of CVD in women giving birth to infants with macrosomia was attenuated after adjusting for cardiometabolic risk factors [29]. Previous studies have shown various results in women who deliver infants with LBW and have a history of stillbirth. In some cohort studies, women who delivered infants with LBW were found to be approximately twice as likely to have CVD in the future [32,33]. In a recent meta-analysis, associations for composite CVD were two-fold for stillbirth, but no association was noted for LBW or small-for-gestational age [34]. Another meta-analysis reported that women with a history of miscarriage and/or stillbirth were more likely to develop CHD, but not stroke, compared with women without a similar history [35]. These various results could be due to the intervention of genetic and racial factors. A study on the association between stillbirth and all-cause mortality according to nationality and race reported that, only in women of North African origin, the adjusted hazard ratio for all-cause mortality after stillbirth had increased significantly, whereas other groups showed no statistically significant differences [36]. Another reason could be that previous studies used different definitions for “a small fetus” and “stillbirth” and had different endpoints of the study, including prevalence and mortality. Further large-scale studies on the relationship between APOs and CVD according to racial and national differences are needed.

Our study has some limitations. The greatest limitation of this study is its cross-sectional design, which does not allow the assessment of the causal pathway underlying the observed relationships. Furthermore, it is possible that there was misclassification and under-reporting of diagnosis because diagnoses of diseases were self-reported by the participants. Additionally, selection bias might have occurred. The subjects were recruited from health examination centers and hospitals located in urban areas of Korea, and only those willing to participate were enrolled; therefore, they might not be entirely representative of the Korean population. This type of screening bias has been observed in many prospective cohort studies. Finally, there was insufficient information on factors that could influence the occurrence of chronic diseases, such as the period from childbirth to the study period, changes in diet or socioeconomic environment from childbirth to the study period, and hormone therapy. No information was available regarding the specific states of pregnancy, including the severity of pregnancy complications, gestational weight gain, and postpartum weight retention. In particular, the gestational age at delivery was not evaluated, which could have potentially made the analysis more thorough. Further studies to determine the association of preterm birth and the development of CVD are necessary. Therefore, although we could adjust for many confounders in the present study, the potential for residual confounding by environmental factors remained. Despite these limitations, our study has several strengths. To the best of our knowledge, this is the only study that has examined the long-term effect of the combination of preeclampsia and GDM on various types of chronic diseases. Moreover, this study had a large sample size, and we were able to adjust for several potential confounders, including family history and social behaviors, and assess their independent and additive effects on the risk of CVD later in life.

In conclusion, APOs moderately influenced the future development of maternal CVD and metabolic derangements, independent of family history and social behaviors. Moreover, the risk of CVD and cardiovascular risk factors did not increase further when preeclampsia occurred in combination with GDM. Further prospective studies are needed to clarify the relationship between APOs and maternal CVD in the future.

## Figures and Tables

**Table 1 jcm-11-01457-t001:** The characteristics of the study population.

	Women without APOs (*n* = 24,910)	Women with APOs (*n* = 5264)	*p*-Value
**Baseline characteristics**			
Age, years	53.31 ± 8.27	51.70 ± 7.56	<0.001
Age of first delivery, years	25.09 ± 2.95	25.65 ± 2.92	<0.001
Age of last delivery, years	29.88 ± 3.69	30.21 ± 3.78	<0.001
**Chronic diseases**			
Hypertension	4537 (18.21)	942 (17.90)	0.586
Diabetes	1267 (5.09)	314 (5.97)	0.009
Hyperlipidemia	1549 (6.22)	381 (7.24)	0.006
Stroke	275 (1.10)	71 (1.35)	0.130
Transient cerebral ischemia	41 (0.16)	6 (0.11)	0.398
Angina pectoris	463 (1.86)	110 (2.09)	0.265
Parkinson’s disease	20 (0.08)	3 (0.06)	0.659
**Family history of chronic diseases**			
Hypertension	6410 (25.73)	1672 (31.76)	<0.001
Diabetes	3737 (15.00)	1011 (19.21)	<0.001
Stroke	3378 (13.56)	848 (16.11)	<0.001
Angina pectoris	1558 (6.25)	415 (7.88)	<0.001
**Behavioral variables**			
Current smoking	581 (2.33)	148 (2.81)	0.040
Current alcohol drinking	7381 (29.63)	1652 (31.38)	0.012
**Lactation history**			
Lactation	22,715 (91.19)	4651 (88.35)	<0.001
Sibling number of lactations	2.55 ± 1.07	2.35 ± 0.91	<0.001
Duration of lactation, m	31.40 ± 28.65	26.52 ± 22.38	<0.001
**Metabolic syndrome and its components ^1^**			
Metabolic syndrome	5018 (20.14)	1023 (19.43)	0.242
Abdominal obesity alone	6710 (26.94)	1455 (27.64)	0.297
Elevated BP alone	9986 (40.09)	2064 (39.21)	0.237
Elevated FBS alone	5752 (23.09)	1314 (24.96)	0.004
High TG alone	5432 (21.81)	1110 (21.09)	0.250
Low HDL-C alone	7452 (29.92)	1519 (28.86)	0.127

Data are presented as mean ± standard error or *n* (%). Abbreviations: APOs, adverse pregnancy outcomes; BP, blood pressure; FBS, fasting blood sugar; TG, total triglyceride; HDL-C, high-density lipoprotein cholesterol. ^1^ Components: abdominal obesity was defined as a waist circumference ≥ 80 cm. Elevated BP was defined as systolic/diastolic pressure ≥ 130/85 mm Hg or drug treatment for hypertension. Elevated FBS was defined as FBS level ≥ 100 mg/dL or treatment for elevated glucose. High TG was defined as TG ≥ 150 mg/dL or medication use. Low HDL-C was defined as an HDL-C level < 50 mg/dL or medication use.

**Table 2 jcm-11-01457-t002:** The body measurements and laboratory data of the study population.

	Women without APOs (*n* = 24,910)	Women with APOs (*n* = 5264)	*p*-Value
**Body measurements**			
Height, cm	155.65 ± 5.28	156.53 ± 5.23	<0.001
Weight, Kg	57.82 ± 7.36	59.17 ± 7.88	<0.001
WC, cm	79.73 ± 8.03	80.05 ± 8.20	0.009
HC, cm	94.37 ± 5.54	94.95 ± 5.74	<0.001
PR	67.82 ± 9.46	67.49 ± 9.61	0.032
Systolic BP, mmHg	121.93 ± 16.42	121.41 ± 16.12	0.035
Diastolic BP, mmHg	75.86 ± 10.09	75.79 ± 10.17	0.630
**Laboratory findings**			
FBS, mg/dL	93.21 ± 19.65	94.62 ± 24.61	<0.001
GGT	21.56 ± 22.67	21.59 ± 21.36	0.917
AST	23.11 ± 14.12	22.90 ± 13.60	0.320
ALT	20.16 ± 17.50	20.31 ± 24.04	0.666
Albumin	4.63 ± 0.28	4.64 ± 0.28	0.010
BUN	13.66 ± 3.78	13.56 ± 3.77	0.058
Creatinine	0.77 ± 0.18	0.77 ± 0.14	0.297
Uric acid	4.25 ± 0.97	4.28 ± 0.96	0.063
TC, mg/dL	199.56 ± 35.51	199.13 ± 35.60	0.428
HDL-C, mg/dL	56.92 ± 12.74	57.14 ± 12.68	0.259
TG, mg/dL	115.60 ± 76.05	113.51 ± 75.66	0.070
hsCRP	0.13 ± 0.37	0.14 ± 0.33	0.784

Data are presented as mean ± standard error or n (%). Abbreviations: WC, waist circumference; HC, hip circumference; PR, pulse rate; BP, blood pressure; FBS, fasting blood sugar; GGT, γ-glutamyl transferase; AST, aspartate aminotransferase; ALT, alanine aminotransferase; BUN, blood urea nitrogen; TC, total cholesterol; HDL-C, high-density lipoprotein cholesterol; TG, total triglyceride; hsCRP, high-sensitivity C-reactive protein.

**Table 3 jcm-11-01457-t003:** Logistic regression analysis for APOs for predicting chronic disease.

	Hypertension	Diabetes Mellitus	Hyperlipidemia	Transient Ischemia	Stroke	Angina Pectoris
	aOR (95% CI)	aOR (95% CI)	aOR (95% CI)	aOR (95% CI)	aOR (95% CI)	aOR (95% CI)
Composite APOs	1.093 (1.004, 1.189)	1.379 (1.207, 1.575)	1.269 (1.126, 1.429)	0.685 (0.289, 1.622)	1.414 (1.083, 1.847)	1.351 (1.090, 1.675)
Preeclampsia	1.394 (1.215, 1.599)	0.988 (0.775, 1.260)	1.194 (0.976, 1.461)	1.324 (0.406, 4.318)	1.313 (0.842, 2.048)	1.429 (1.012, 2.019)
GDM	1.417 (0.974, 2.061)	6.640 (4.549, 9.693)	1.665 (1.024, 2.707)	-	1.092 (0.266, 4.477)	0.671 (0.164, 2.739)
Stillbirth	0.944 (0.680, 1.310)	0.953 (0.563, 1.613)	0.725 (0.426, 1.234)	-	1.597 (0.700, 3.645)	0.753 (0.306, 1.851)
Macrosomia	0.881 (0.780, 0.995)	1.573 (1.325, 1.867)	1.273 (1.083, 1.496)	0.250 (0.034, 1.817)	1.434 (1.000, 2.058)	1.292 (0.959, 1.741)
LBW	1.093 (0.938, 1.273)	0.884 (0.667, 1.171)	1.200 (0.966, 1.491)	1.405 (0.431, 4.582)	1.383 (0.862, 2.218)	1.276 (0.863, 1.886)
Preeclampsia only	1.409 (1.227, 1.618)	1.111 (0.869, 1.422)	1.290 (1.055, 1.578)	1.303 (0.403, 4.214)	1.446 (0.933, 2.244)	1.466 (1.036, 2.074)
GDM only	1.593 (1.011, 2.511)	9.069 (5.938, 13.851)	2.496 (1.484, 4.198)		1.918 (0.469, 7.854)	0.566 (0.078, 4.085)
Preeclampsia + GDM	1.579 (0.834, 2.991)	3.287 (1.473, 7.338)	0.559 (0.136, 2.298)			1.263 (0.173, 9.224)

Adjusted by age, family history (hypertension, diabetes, angina, stroke), present smoking, and present alcohol. Abbreviations: aOR, adjusted odds ratio; CI, confidence interval; APOs, adverse pregnancy outcome; GDM, gestational diabetes; LBW, low birth weight. The unadjusted basic model is described in Appendix A.

**Table 4 jcm-11-01457-t004:** Logistic regression analysis for APOs for predicting metabolic syndrome and its components.

	Metabolic Syndrome	Abdominal Obesity	Elevated BP	Elevated FBS	Low HCL-C	High TG
	aOR (95% CI)	aOR (95% CI)	aOR (95% CI)	aOR (95% CI)	aOR (95% CI)	aOR (95% CI)
Composite APOs	1.104 (1.020, 1.194)	1.172 (1.094, 1.255)	1.074 (1.006, 1.147)	1.193 (1.111, 1.280)	1.009 (0.944, 1.078)	1.038 (0.964, 1.118)
Preeclampsia	1.115 (0.974, 1.276)	1.185 (1.051, 1.335)	1.349 (1.204, 1.510)	1.040 (0.916, 1.179)	0.939 (0.835, 1.057)	1.001 (0.879, 1.141)
GDM	1.814 (1.309, 2.515)	1.300 (0.959, 1.763)	1.372 (1.034, 1.821)	2.512 (1.907, 3.310)	1.210 (0.908, 1.613)	1.281 (0.929, 1.766)
Stillbirth	1.015 (0.747, 1.379)	1.025 (0.774, 1.356)	1.133 (0.862, 1.488)	1.029 (0.770, 1.376)	0.969 (0.735, 1.276)	1.111 (0.830, 1.487)
Macrosomia	1.098 (0.986, 1.223)	1.350 (1.231, 1.479)	0.888 (0.811, 0.972)	1.297 (1.180, 1.426)	1.042 (0.952, 1.141)	1.001 (0.903, 1.109)
LBW	0.956 (0.824, 1.108)	0.824 (0.721, 0.941)	1.060 (0.942, 1.193)	0.926 (0.809, 1.059)	0.973 (0.862, 1.099)	1.090 (0.954, 1.245)
Preeclampsia only	1.124 (0.980, 1.289)	1.176 (1.042, 1.328)	1.367 (1.219, 1.532)	1.076 (0.947, 1.223)	0.955 (0.847, 1.076)	1.002 (0.878, 1.144)
GDM only	1.908 (1.286, 2.831)	1.209 (0.827, 1.765)	1.511 (1.075, 2.125)	3.001 (2.167, 4.157)	1.396 (0.997, 1.954)	1.207 (0.813, 1.791)
Preeclampsia + GDM	1.866 (1.064, 3.270)	1.817 (1.104, 2.991)	1.532 (0.940, 2.494)	1.848 (1.121, 3.048)	0.810 (0.469, 1.397)	1.468 (0.857, 2.516)

Adjusted by age, family history (hypertension, diabetes, angina, stroke), present smoking, and present alcohol. Abbreviations: BP, blood pressure; FBS, fasting blood sugar; HDL-C, high-density lipoprotein cholesterol; TG, triglyceride; aOR, adjusted odds ratio; CI, confidence interval; APO, adverse pregnancy outcome; GDM, gestational diabetes; LBW, low birth weight. The unadjusted basic model is described in Appendix A.

## Data Availability

The data in this article are obtained from public databases that are open and transparent.

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
