# Peer review of "Adverse Pregnancy Outcomes and Maternal Chronic Diseases in the Future: A Cross-Sectional Study Using KoGES-HEXA Data"

_jcm, 2022, doi:10.3390/jcm11051457_

Round 1

Reviewer 1 Report

Thanks to let me review this paper. In my opinion even if the subject dealt with is not new, the amount of data presented gives value to the manuscript. I have some observations: 

  • Introduction: the incidence and prevalence of cardiovascular disease in women is not specified. How many have APO in pregnancy?
  • Materials and methods: it could be more precise in the definition of each APOs, or at least mention the one to which reference is made; in family history to what degree of kinship has been investigated?
  • Discussion: among the limiting factors add the gestational age ( if possible), which would have made the anamnesis more complete and perhaps would have allowed an additional analysis (eg correlation between preterm birth and the development of CVD)

Author Response

  • Introduction: the incidence and prevalence of cardiovascular disease in women is not specified. How many have APO in pregnancy?
    • We added the prevalence of CVD and APOs (line 36-38, 54-55)

  • Materials and methods: it could be more precise in the definition of each APOs, or at least mention the one to which reference is made; in family history to what degree of kinship has been investigated?
    • We included the definition of APOs (line 102-104). Family history was limited to siblings and parents (line 107).

  • Discussion: among the limiting factors add the gestational age ( if possible), which would have made the anamnesis more complete and perhaps would have allowed an additional analysis (eg correlation between preterm birth and the development of CVD)
    • We added the gestational age of delivery as limiting factor (line 297-300)

Reviewer 2 Report

The objective of this work was to elucidate the most relevant aspects between adverse pregnancy outcomes (APOs) and the risk of chronic diseases, including cardiovascular disease (CVD) and metabolic syndrome (MS), in the future. The authors designed a large scale cohort study to evaluate the influence of APOs (preeclampsia, gestational diabetes mellitus [GDM], stillbirth, macrosomia, and low birth weight) on the incidence of chronic diseases. To do this, they used health examinee data from the Korean Genome and Epidemiology Study 20 (KoGES-HEXA) and extracted data of parous women (n = 30,174; mean age, 53.02 years) for the analysis. The authors found that preeclampsia and GDM were associated with an increased risk of some chronic diseases; however, the combination of preeclampsia and GDM did not have an additive effect on the risk. APOs moderately influenced the future development of maternal CVD and metabolic derangements, independent of family history and social behaviors.

The manuscript is well written and easy to read and understand, and has important strengths, including the large sample size used and the number and importance of the adjusted variables. However, it also has important limitations such as:

1) Cross-sectional design, which does not allow the establishment of a phenomenon of causality, but only of association between different variables.

2) The information collection method (self-reported) that favors the appearance of information biases with the consequent errors of diagnosis and classification.

3) The inclusion criteria in the study that were not rigorous enough and that should be explained in more detail in the Methodology section.

4) Insufficient information on factors that could influence the occurrence of chronic diseases (changes in diet or socioeconomic environment, therapies during pregnancy…etc).

5) Insufficient information about pregnancy complications.

6) Surprisingly, the combination of preeclampsia and GDM did not have an additive effect on the risk. This finding should be further analyzed and the possible causes and/or mechanisms involved discussed.

Author Response

  • Cross-sectional design, which does not allow the establishment of a phenomenon of causality, but only of association between different variables.

-> We understand it is the limitation, therefore we expressed the sentence in discussion (line284-286).

2) The information collection method (self-reported) that favors the appearance of information biases with the consequent errors of diagnosis and classification.

-> We understand it is the limitation, therefore we expressed the sentence in discussion (line286-288).

3) The inclusion criteria in the study that were not rigorous enough and that should be explained in more detail in the Methodology section.

-> We added more explanation of inclusion criteria and reference (line 83-88).

4) Insufficient information on factors that could influence the occurrence of chronic diseases (changes in diet or socioeconomic environment, therapies during pregnancy…etc).
-> We understand it is the limitation, therefore we expressed the sentence in discussion (line292-295).

5) Insufficient information about pregnancy complications.

-> We understand it is the limitation, therefore we expressed the sentence in discussion (line295-297).

6) Surprisingly, the combination of preeclampsia and GDM did not have an additive effect on the risk. This finding should be further analyzed and the possible causes and/or mechanisms involved discussed.

 -> We understand it is the limitation, therefore we added the limitation (line259-261).

Round 2

Reviewer 2 Report

I am satisfied with the authors' responses to my comments and with the changes made to the new manuscript.